# Treatment of Prolactinoma

**DOI:** 10.3390/medicina58081095

**Published:** 2022-08-13

**Authors:** Warrick J. Inder, Christina Jang

**Affiliations:** 1Department of Diabetes and Endocrinology, Princess Alexandra Hospital, Woolloongabba 4102, Australia; 2Academy for Medical Education, Faculty of Medicine, The University of Queensland, Herston 4029, Australia; 3Department of Endocrinology and Diabetes, Royal Brisbane and Women’s Hospital, Herston 4029, Australia; 4Faculty of Medicine, The University of Queensland, Herston 4029, Australia

**Keywords:** prolactinoma, dopamine agonist, transsphenoidal surgery, temozolomide, stereotactic radiosurgery

## Abstract

Prolactinomas are the commonest form of pituitary neuroendocrine tumor (PitNET), representing approximately half of such tumors. Dopamine agonists (DAs) have traditionally been the primary treatment for the majority of prolactinomas, with surgery considered the second line. The aim of this review is to examine the historical and modern management of prolactinomas, including medical therapy with DAs, transsphenoidal surgery, and multimodality therapy for the treatment of aggressive prolactinomas and metastatic PitNETs, with an emphasis on the efficacy, safety, and future directions of current therapeutic modalities. DAs have been the mainstay of prolactinoma management since the 1970s, initially with bromocriptine and more recently with cabergoline. Cabergoline normalizes prolactin in up to 85% of patients and causes tumor shrinkage in up to 80%. Primary surgical resection of microprolactinomas and enclosed macroprolactinomas performed by experienced pituitary neurosurgeons have similar remission rates to cabergoline. Aggressive prolactinomas and metastatic PitNETS should receive multimodality therapy including high dose cabergoline, surgery, radiation therapy (preferably using stereotactic radiosurgery where suitable), and temozolomide. DAs remain a reliable mode of therapy for most prolactinomas but results from transsphenoidal surgery in expert hands have improved considerably over the last one to two decades. Surgery should be strongly considered as primary therapy, particularly in the setting of microprolactinomas, non-invasive macroprolactinomas, or prior to attempting pregnancy, and has an important role in the management of DA resistant and aggressive prolactinomas.

## 1. Introduction

Prolactin is a 199 amino acid, 23 kDa protein secreted by pituitary lactotroph cells which are derived from the Pit1 lineage. Prolactin was not isolated in humans until the early 1970s, when Kleinberg and Frantz developed a bioassay that was able to quantify elevations in prolactin among breastfeeding women and patients with galactorrhea [1].

Hypothalamic dopamine is the main regulator of prolactin secretion. While other anterior pituitary hormones are predominantly under positive hypothalamic regulation, dopamine is inhibitory. While experimental evidence has shown a number of hormones such as thyrotrophin releasing hormone (TRH) [2], vasoactive intestinal polypeptide (VIP) [3], and oxytocin may increase prolactin; to date, none of these or indeed other putative candidate hormones fulfil criteria for being a genuine hypothalamic prolactin releasing factor [4]. Prolactin dynamics do not appear to be explained by the effect of dopamine alone, however, but the identity of a prolactin releasing factor remains elusive [4].

There are several physiological causes of hyperprolactinemia of which the commonest are pregnancy [5] and breastfeeding [6]. An increase in prolactin may be brought about by nipple [7] and thoracic stimulation [8] and spinal cord injury [9]. Prolactin may be elevated after seizures [10], including electro-convulsive therapy [11]. Prolactin is also stimulated during orgasm in both sexes [12].

Prolactinomas are the commonest pituitary neuroendocrine tumor (PitNET), with a recent epidemiological review estimating that this tumor subtype makes up 53% of PitNETs [13]. The aim of this review is to examine the historical and modern management of prolactinomas, with an emphasis on the efficacy, safety, and future directions of current therapeutic modalities.

## 2. Diagnosis of Prolactinoma

The pathway to diagnosis of prolactinoma may be via hormonal investigation of symptoms of reproductive dysfunction such as menstrual disturbance and galactorrhea in females or erectile, dysfunction, loss of libido, and rarely galactorrhea in males. Alternatively, it may be via the discovery of a PitNET on neuroimaging, either as an incidental finding for an unrelated indication or to investigate pituitary mass effect symptoms such as a bitemporal visual field defect, followed by a secondary measurement of prolactin.

Prolactin is easily measured via immunoassay, with many laboratories using sex-based reference ranges, with a lower upper limit of normal for males. While the precise upper limit is assay-specific, it is usually in the order of 400–500 mIU/L (18.8–23.5 µg/L). The likelihood of a pathological cause increasing once the concentration is more than twice the upper limit of normal, or approximately 1000 mIU/L (47 µg/L). The stress of venepuncture may induce an acute rise in prolactin, which can be averted by sampling through an intravenous cannula. In one study which sampled at 30 min intervals for 2 h, 61% of women with hyperprolactinemia on two previous independent samples obtained via venepuncture had normal serum prolactin [14]. A macroprolactinoma is usually associated with a serum prolactin of >10-fold the upper limit of the reference range or 4000–5000 mIU/L (188–235 µg/L). Any sellar or parasellar lesion which interrupts the transport of dopamine from the hypothalamus to the anterior pituitary, including non-functioning sellar masses may result in hyperprolactinemia—the so-called “stalk effect”. Most non-functioning masses result in a serum prolactin of <2000 mIU/L (94 µg/L) [15], though exceptions to this rule can occur [16]. Extremely high prolactin concentrations may result in falsely low readings when using two site immuno- or chemiluminometric assays due to saturation of both antibodies—known as the “hook effect”. If in doubt, serial dilution to 1:100 should be undertaken to discern the true prolactin concentration [17]. Uncommonly, PitNETs of Pit-1 lineage expressing prolactin (often plurihormonal), may be non-secretory or “poorly” functioning with serum prolactin in the range expected for stalk effect <2000 mIU/L (94 µg/L). It may be impossible to distinguish such lesions from non-functioning PitNETs of a different lineage without histological confirmation including transcription factor immunohistochemistry [18,19]. Finally, clinicians must be aware of drug-induced hyperprolactinemia (particularly with antipsychotics [20] and antiemetics [21]) and the presence of macroprolactin, a biologically inactive immune complex between prolactin and IgG [22] which is causative in approximately 10% of cases of hyperprolactinemia [23].

As with all PitNETs, a tumor < 10 mm in diameter is defined as a microadenoma, ≥10 mm diameter a macroadenoma, while a “giant” macroadenoma has a maximal diameter of >40 mm. Microprolactinomas are particularly common in young women, and hyperprolactinemia is the commonest pituitary cause of secondary amenorrhea [24]. Natural history studies of patients with microprolactinomas have demonstrated that they rarely increase in size, even if left untreated [25].

Prolactinomas are less common in men overall, but the proportion of macroadenomas, particularly large, invasive tumors is higher [26]. Reasons for the sex-related size difference are only partly related to the prominence of galactorrhea and amenorrhea in women as an obvious signal of reproductive dysfunction [26]. As noted above, microprolactinomas rarely increase in size and there is evidence that prolactinomas in men are innately more aggressive with a greater propensity to growth [27] and a higher rate of dopamine agonist resistance [28].

Prolactinomas may present as part of a genetic syndrome such as multiple endocrine neoplasia type 1 (MEN1) [29] or type 4 (MEN4) [30], Succinate dehydrogenase (SDHx) mutations [31], and aryl hydrocarbon receptor interacting protein (AIP)-associated familial pituitary adenoma syndrome (FIPA) [32]. While a detailed examination of this topic is beyond the scope of this review, the diagnosis of prolactinoma should always be accompanied by detailed family history and genetic testing initiated according to recent recommendations [33].

## 3. Management

### 3.1. Dopamine Agonists

Shortly after the isolation of prolactin in the early 1970s, bromocriptine was found to inhibit prolactin secretion and began to be used in the treatment of galactorrhea and associated hypogonadism [34]. By the end of the 1970s with the development of computed tomography, it was demonstrated by several investigators that the dopamine agonist (DA) bromocriptine resulted in regression of macroprolactinomas in addition to reducing serum prolactin concentrations [35,36].

Upon diagnosis of prolactinoma, the primary approach has been the use of medical therapy in the form of DAs. Bromocriptine was the treatment of choice until the emergence of cabergoline in the late 1980s-early 1990s [37]. A comparative trial between the two agents for the treatment of hyperprolactinemic amenorrhea was published in 1994 [38]. In this study, cabergoline was superior in efficacy in restoring normoprolactinemia and ovulatory menstrual cycles, required less frequent dosing, and had fewer adverse gastrointestinal effects [38]. Another DA, quinagolide, was also developed during the 1990s [39,40]. It has the advantage of not being an ergotamine derivative and is better tolerated than bromocriptine in some patients [40]. Overall, cabergoline has shown superior efficacy, with evidence that prolactinomas that are resistant to bromocriptine or quinagolide may respond to cabergoline treatment, resulting in both improved biochemical control and greater tumor shrinkage [41]. A large comparative study examining patients previously treated with other DAs (bromocriptine or quinagolide) compared to DA-naïve patients showed that the drug-naïve patients had greater tumor shrinkage on lower cabergoline doses compared to those previously treated including those who were DA responsive, intolerant or resistant [42]. In the treatment of macroprolactinomas, cabergoline was superior to the other dopamine agonists both in terms of prolactin normalization and tumor shrinkage [43]. A review from Brazil demonstrated that cabergoline induced prolactin normalization in 86% of treatment naïve prolactinoma patients overall (91% in patients with microadenoma, 83% in macroadenoma) and resulted in tumor shrinkage in 80% of treatment naïve macroprolactinomas [44].

As previously stated, men tend to have larger, more resistant prolactinomas than women and specific studies examining the efficacy of cabergoline in male patients emerged demonstrating the effectiveness of cabergoline in terms of return of sexual function and semen analysis [45].

#### Dose and Administration

Typically, cabergoline is commenced at low doses of 0.25 mg twice weekly, taken in the evening after food to minimize gastrointestinal side effects (predominantly nausea). A standard microadenoma dose would be 0.5–1 mg per week, with up-titration indicated if this dose did not result in normalization of the serum prolactin within 6–8 weeks. For macroprolactinomas, a similar initial dose may be used; with the potential up-titration of the dose to 1 mg twice weekly by week three. Where visual field defects have resulted from suprasellar extension, the dose can be increased more rapidly to 1 mg twice weekly by the start of week two. Failure of prolactin to normalize on cabergoline 2 mg/week has been used to define a resistant prolactinoma [46]—dose escalation up to 12 mg weekly has been reported [47] (see Section 3.6). Repeat visual field assessment after two weeks of therapy may already demonstrate normalization in cabergoline-sensitive subjects. In women, vaginal administration of cabergoline can be used to minimize gastrointestinal side effects should they occur [48].

Quinagolide is commenced at a dose of 37.5–75 µg daily, and up-titrated as indicated depending on the prolactin response. The usual dose range is 75–450 µg daily [49], but doses of up to 600 µg daily have been used [50].

Bromocriptine is generally commenced at a dose of 1.25–2.5 mg daily, with up-titration to twice daily dosing as required. Doses of up to 30 mg/day in 2 divided doses may be used. In practice, bromocriptine is rarely used these days, although some experts still advise its use in women seeking pregnancy because its longer history of use has provided greater case numbers to confirm safety during pregnancy (see below) [51].

### 3.2. Adverse Effects of Dopamine Agonists

#### 3.2.1. Common Gastrointestinal and Neurological Side Effects

Dopamine agonists have several treatment-limiting side effects, including nausea, vomiting, fatigue, headache, and dizziness. The study by Webster et al. showed that 50% of women on bromocriptine had some degree of nausea compared to 31% with cabergoline. Severe vomiting occurred in 5% of women on bromocriptine compared to 0% on cabergoline. The rate of headache and dizziness were similar at around 30% and 25% respectively for both medications. Overall treatment withdrawal occurred in 12% of bromocriptine treated patients compared to 3% in the cabergoline group [38].

#### 3.2.2. Cabergoline-Associated Valvulopathy

In 2007, two studies published data on an increased risk of specific cardiac valvulopathy associated with high dose cabergoline treatment in Parkinson’s disease [52,53]. In these studies, the cumulative cabergoline dose which resulted in these valvular abnormalities was several-fold higher than what is generally used in the treatment of prolactinomas. Subsequent studies have attempted to define the risk in prolactinoma patients. Recommendations to perform regular echocardiograms in prolactinoma patients on cabergoline to monitor the cardiac valves were released before this risk had been carefully evaluated. There have now been numerous studies on the rate of cardiac valve abnormalities in prolactinoma patients taking cabergoline, but many have reported non-specific valve pathology rather than the specific features which define cabergoline-associated valvulopathy (CAV)—namely: 1. Moderate or severe valvular regurgitation, 2. Valve thickening and 3. Valve restriction [54]. The absence of calcification and myxomatous change separates CAV from age-related sclerosis and myxomatous change respectively. Using these strict criteria, there have only been three definite published cases of CAV in patients with prolactinoma [55]. A recent study from primary care in the UK by authors who were instrumental in the development of the UK echocardiogram screening guidelines [56], showed no evidence of an increase in CAV in patients treated with cabergoline for prolactinoma [57]. The UK guidelines regarding the recommended frequency of screening echocardiograms advocate a routine echocardiogram at baseline, but this would have an extremely low yield in people with a normal cardiovascular examination, particularly in young women with microadenomas. We suggest following the recommendations of Caputo et al. and undertaking annual clinical cardiovascular examination and reserve echocardiograms for those who (a) have an audible cardiac murmur on clinical examination, (b) have been treated on a cabergoline dose of >3 mg/week for 5 years or longer or (c) are over 50 years of age (and are more likely to have other cardiac valve pathology) [54]. This strategy successfully detected the third published case of CAV in prolactinoma [55].

#### 3.2.3. Impulse Control Disorders

Impulse control disorders (ICDs) were previously thought to be rare in patients treated with dopamine agonists for prolactinoma, but a number of studies across the last 10 years have highlighted their prevalence (8–25%) and the adverse consequences that may arise [58]. ICDs are characterized by the failure to resist impulses to engage in a pleasurable but harmful activity [59]. ICDs associated with DA use include hypersexuality, pathological gambling, compulsive buying, punding (a preoccupation with meaningless motor activities such as rearranging objects or cleaning), preoccupation with hobbies, and purposeless wandering either on foot or in a motor vehicle [60]. Identified risk factors for ICD are male sex, a eugonadal state in both sexes, a lower Hardy’s tumor score (less invasive tumor), and the presence of psychiatric comorbidities such as anxiety and depression for hypersexuality and age for compulsive buying [61]. ICD is associated with bromocriptine, quinagolide, and cabergoline, with no good evidence to suggest that changing DA will alleviate symptoms [62]. Using a series of neuropsychological questionnaires in an unselected group of patients with DA-treated hyperprolactinemia, 61% were positive for any ICD [61]. In a specific questionnaire examining hypersexuality, 8% of DA treated patients were positive compared to 0% of the controls [61]. Neither the dose nor duration of DA therapy are predictive of the development of ICD [61,63]. While occasional patients may improve with dose reduction [62,64], this is not usually the case, and often resolution of the ICD requires cessation of the DA.

#### 3.2.4. Psychosis

Dopamine is well recognized as a critical mediator of psychosis, and dopamine antagonists are frequently used as antipsychotics [20,65]. The management options for people with prolactinomas and underlying psychiatric conditions such as schizophrenia are discussed in a later section. However, a new onset of psychosis in a patient without any previous history of psychiatric disease may rarely occur following the initiation of DAs. Turner et al. described 8 cases of new onset psychosis out of a cohort of 600 patients (1.3%) treated with bromocriptine or lisuride [66]. In a 1995 review, Boyd examined the relationship between bromocriptine and psychosis, uncovering 62 patients reported in the literature at that time who had developed psychosis, being treated for a range of indications including Parkinson’s disease, acromegaly, and prolactinoma [65]. Contemporary series examining the rate of psychosis in patients treated with cabergoline or quinagolide are lacking and evidence is confined to isolated case reports [67,68,69].

#### 3.2.5. Effect on Future Pituitary Surgery

DA therapy, particularly bromocriptine, may induce fibrosis in prolactinomas [70]. This has been highlighted as a factor in non-remission in some surgical studies [70,71]. This effect is considerably less frequent in patients treated with cabergoline [70,72], and other studies have not found a negative impact of pre-surgical DA use on post-operative remission [73,74]. A 2021 meta-analysis, however, did show a significant negative effect of DA pre-treatment [75].

#### 3.2.6. Cerebrospinal Fluid Rhinorrhea

Large, invasive macroprolactinomas can invade and erode the skull base, with the tumor mass essentially plugging the defect. Leakage of cerebrospinal fluid (CSF) may occur with the development of rhinorrhea, which in turn results in a major risk of serious infection (meningitis, encephalitis) and pneumoencephaly [76]. The incidence of CSF rhinorrhea in macroprolactinoma patients has been reported at 8.7% of cases [77]. In this study, 30% of the cases were spontaneous, and 70% were induced by dopamine agonists reducing the size of the “plug” [77]. The occurrence of DA-related CSF rhinorrhea requires surgical repair of the defect in approximately 90% of cases [78].

### 3.3. Special Situations

#### 3.3.1. Pregnancy

With the peak incidence of prolactinomas occurring during the reproductive years, managing prolactinomas in pregnancy is a frequently encountered clinical scenario. In normal pregnancy physiology, the normal pituitary enlarges by 30–40% due to lactotroph hypertrophy in response to increased circulating estrogen [79]. Circulating prolactin levels progressively increase such that by the third trimester, levels will be up to ten-fold higher than non-pregnancy [5,80]. In the presence of a pituitary tumor, a mass effect may lead to headaches and or compression of the optic chiasm. The management of prolactinomas in pregnancy depends on the size and characteristics of the tumor.

In most cases, the diagnosis of a prolactinoma will be made prior to conception. Women with hyperprolactinemia present with oligo-amenorrhea and subfertility. Elevated prolactin levels suppress gonadotrophin secretion, leading to anovulatory cycles [81]. Furthermore, hyperprolactinemia may lead to luteal phase insufficiency by interfering with normal progesterone synthesis by the corpus luteum [82,83]. Dopamine agonists remain the mainstay of treatment for prolactinomas in women attempting pregnancy, effectively restoring ovulation in 80–90% of cases [84].

The likelihood of significant microprolactinoma enlargement during pregnancy is small—in the order of 2–3% [85]. Macroprolactinomas on the other hand have been reported to undergo symptomatic enlargement in 21% of cases, but as low as 4.7% with prior treatment [85]. In the case of intrasellar macroprolactinomas, it is estimated that this risk is similar to or only marginally higher than microprolactinomas [85].

The use of bromocriptine in pregnancy was described in the 1970′s [86]. Since then, reports of over 6000 pregnancies induced by bromocriptine have demonstrated no adverse outcomes [87,88]. A study of 64 children born to 53 mothers treated with bromocriptine for part of the pregnancy is also reassuring [89]. Included were 23 children whose mothers were treated for 30 weeks or more and after a follow-up of up to 9 years, no abnormalities were reported, including psychological development.

Initial reports of cabergoline use in pregnancy were published in 1994 [90]. Multiple reports of pregnancies conceived while taking cabergoline have also shown no effect on pregnancy outcomes compared to the normal population [87]. In a retrospective study of 100 pregnancies conceived while on treatment with cabergoline, there were no differences in pregnancy outcomes or congenital malformations when compared to a control population [91]. A more recent study included 25 women with macroprolactinomas who continued cabergoline therapy throughout pregnancy with cumulative exposure of 52.1 ± 42.4 mg, compared to a group of women whose therapy was ceased at the time of conception (mean cumulative dose 14.1 ± 14.1 mg) [92]. There were no differences in neonatal malformations or spontaneous miscarriage between the two groups [92]. Cabergoline is now recommended as a first line treatment because of its better tolerability and superior efficacy [93].

Quinagolide should not be used in pregnancy. Based on manufacturer data of 176 pregnancies in 157 women in which quinagolide was continued into pregnancy for a median duration of 37 days, 24 spontaneous miscarriages, 1 ectopic pregnancy, and 1 still birth at 31 weeks were observed [93,94].

For patients with microprolactinomas and intrasellar macroadenomas, it is currently recommended that treatment with dopamine agonists and therapy be withdrawn once pregnancy is confirmed [51]. Patients should be assessed every 3 months noting headaches or visual disturbance [95]. For macroprolactinomas, the most common practice is to cease dopamine agonist therapy when pregnancy is confirmed. Ideally, the effectiveness of dopamine agonist in tumor shrinkage should be demonstrated before pregnancy, aiming to reduce the cranio-caudal dimension to within the sellar boundary [51]. Once pregnant, it is recommended that patients undergo clinical examination every month and visual field testing performed every 3 months [96]. It is controversial whether prolactin levels should be measured in pregnancy, with some experts suggesting a role for measurement only in the case of macroprolactinomas with comparison to pregnancy specific ranges [95]. However, the European Clinical Practice Guidelines on functioning and non-functioning pituitary adenomas in pregnancy recommend not measuring prolactin during pregnancy [93].

Patients with new onset or worsening headaches should have a visual field examination and MRI without gadolinium. If there is evidence of prolactinoma enlargement, dopamine agonists should be recommenced. Surgery should be considered if medical therapy is unsuccessful or if there is evidence of apoplexy with visual deterioration [93]. An alternative approach to suprasellar macroadenomas is surgical debulking of the tumor before pregnancy, particularly for patients where dopamine agonists have shown limited efficacy on tumor shrinkage [51,85]. Treatment options should be presented to women as part of pre-conception counselling.

#### 3.3.2. Psychiatric Illness

Hyperprolactinemia is a frequent consequence of antipsychotic medication [20], but not all cases of hyperprolactinemia are purely drug-induced and co-existing prolactinomas do occur. In this setting, perhaps somewhat surprisingly given that DAs may cause psychosis, the use of cabergoline in combination with continuation of the antipsychotic medication rarely results in exacerbation of the underlying psychiatric condition [97,98]. An alternative is to use the partial dopamine agonist antipsychotic agent aripiprazole. There are now several reports of prolactinomas being successfully treated with aripiprazole, either as monotherapy or as an add-in to existing antipsychotics [99,100].

### 3.4. Dopamine Agonist Withdrawal

A landmark study published by Colao et al. in 2003 showed that it was possible in some patients with prolactinomas to induce a lasting period of biochemical and radiological remission after cabergoline withdrawal [101]. The presence of a tumor remnant was a significant risk factor for recurrence compared to those patients with no visible tumor [101].

Subsequent longer-term follow-up enabled further characterization of the features which predict remission after cabergoline withdrawal [102]. In this study, all participants had to have a normal serum prolactin at a cabergoline dose of 0.5 mg per week and had a mean duration of treatment of over three years. Maximal tumor diameter at the time of withdrawal was 5.6 mm (mean 1.2 mm) in the microadenoma group and 9.5 mm (mean 2.4 mm) in the macroadenoma group. Follow-up was 24–96 months after cabergoline withdrawal. Recurrence of hyperprolactinemia was associated with both nadir serum prolactin and nadir tumor diameter immediately prior to cabergoline withdrawal, and tumor diameter at diagnosis [102]. The optimal thresholds were a serum prolactin at withdrawal of ≤162 mIU/L and maximal tumor diameter at withdrawal of ≤3.1 mm. Recurrence was seen in all but one patient whose tumor was >6 mm in maximal diameter. Among patients with microprolactinomas who fulfilled both these criteria, remission was maintained in 50/56 (89.3%) while in macroadenomas fulfilling both criteria, 24/33 (72.7%) remained in remission [102].

The 2011 Endocrine Society Hyperprolactinemia Guidelines suggest that dopamine agonist discontinuation should be considered after treatment for at least 2 years in patients with a normal serum prolactin and no visible tumor remnant on imaging [96].

A more recent position statement published by the Italian Association of Clinical Endocrinologists and the International Chapter of Clinical Endocrinology (AME/ICCE) recommends lifelong treatment for men with microprolactinomas, but routine DA withdrawal in women once they reach menopause [103]. They state that DA withdrawal in patients with macroprolactinomas is only occasionally successful. Follow-up of serum prolactin is recommended at 3 months post-withdrawal, with the frequency of further monitoring dependent on this initial result [103].

### 3.5. Surgery

Given the relative ease of medical management for the majority of people with prolactinomas, surgery has until recently been largely viewed as second line therapy for patients that were intolerant or resistant to dopamine agonists [76]. Other circumstances where surgery had a role included the setting of cerebrospinal fluid rhinorrhea, in some instances of pituitary apoplexy, cystic lesions, and to both treat and obtain histological confirmation of whether a macroadenoma with modest hyperprolactinemia is a poorly functioning macroprolactinoma or a non-functioning PitNET with stalk effect [58]. Early studies reporting outcomes of surgically treated prolactinomas yielded discouraging results compared to medical therapy. For example, in the mid-1990s Soule et al. described 34 patients (23 macroadenomas) who underwent pituitary surgery undertaken by a single pituitary neurosurgeon, mostly in the setting of previous dopamine agonist use [104]. Normal post-operative serum prolactin was achieved in only 17.4% of patients with macroprolactinomas and 45.5% of microadenomas [104].

Endoscopic pituitary surgery was first described in 1997 using rigid endoscopes, rapidly progressing from a sublabial to an endonasal approach [105,106]. Across the 2000s, there have been refinements in pituitary surgical techniques and the realization of improved remission rates for prolactinomas, achieved by specialized pituitary surgeons with a high case load. In their 2016 review, Tampourlou et al. reported a remission rate in 81–100% of microprolactinoma patients operated on using the endoscopic technique compared to 71–93% using the microscopic technique [107]. In a later review, remission in patients operated on for microprolactinomas was achieved in 91% of cases managed in high volume centers compared to 77% in low volume centers [108]. These data indicate that with an experienced pituitary neurosurgeon, the remission rate for microprolactinomas with transsphenoidal surgery is now similar to that achieved with DAs [76].

The remission rates do fall in association with increasing tumor size and invasiveness, while female sex and absence of previous DA treatment are associated with increased remission rates [75]. Ikeda et al. examined 138 female patients who underwent surgery for prolactinoma [73]. The remission rate was 86% in microadenomas compared to 74% in macroadenomas. Cavernous sinus invasion was found in 37/138 prolactinomas—the remission rate was 24% in the invasive tumors compared to 94% of the enclosed tumors [73].

Recurrence rates may also relate to tumor size and degree of extension. In a series of over 200 prolactinoma patients surgically managed, recurrence occurred in 7.1% of microprolactinomas but in one third of patients who had suprasellar macroprolactinoma with visual field defects [109]. Overall, the meta-analysis by Wright et al. did not demonstrate any significant factors to predict recurrence due to insufficient data [75].

Another important consideration is the surgical complication rate, particularly given the most common demographic are young women with microprolactinomas where fertility preservation is paramount. Again, complication rates are very uncommon in experienced surgical centers [76]. Tampourlou et al. reported mortality, visual deterioration, and other neurosurgical complications all at 0% in patients who had undergone endoscopic transsphenoidal surgery for microprolactinomas, and new onset pituitary dysfunction ranged between 0–6% with no cases of diabetes insipidus [107]. In the large series from Kreutzer et al., 156 patients had macroadenomas (including 10 giant adenomas) and 56 had microadenomas, with follow-up data on 171 patients (125 macroadenomas, 46 microadenomas) [109]. New anterior pituitary dysfunction occurred in 12 patients (7.0% of the total cohort), all in macro- or giant adenomas (9.6%) with no new permanent diabetes insipidus [109].

Overall, the recent surgical literature highlights improved outcomes in patients with prolactinomas, and in particular the importance of an expert pituitary surgeon operating in a high volume center.

#### 3.5.1. Suggested Indications for Surgery

Given the efficacy and safety of selective transsphenoidal adenomectomy for the treatment of prolactinoma, it is appropriate to consider surgery as a first line treatment option under certain circumstances. The important caveat is the availability of a suitably expert, high volume pituitary neurosurgeon, preferably in a Pituitary Surgery Centre of Excellence.

Microprolactinomas: The option of surgery as first line treatment should be offered to patients presenting with a newly diagnosed, well circumscribed microprolactinoma [76]. As outlined, the remission rates are similar to DA therapy and present a chance of genuine cure with a low risk of post-operative pituitary dysfunction. Patient preference for surgery was recognized as an indication for surgery in the Pituitary Society 2006 Guidelines [110].

Circumscribed macroprolactinomas in females: Surgical outcomes for female patients with non-invasive macroprolactinomas are similar to those of microprolactinomas [73].

Desire for pregnancy: Surgery offers a high probability of remission in microadenomas, thereby eliminating the need for DA therapy to induce ovulation. For women with macroadenomas seeking fertility, debulking surgery reduces the risk of symptomatic tumor growth during a subsequent pregnancy [51].

Cystic prolactinomas: While many cystic prolactinomas may respond well to dopamine agonists [111], surgery is indicated for those cystic tumors which do not [112].

Debulking in DA-resistant prolactinomas, aggressive prolactinomas, and carcinomas: Surgery is considered one of the mainstays of treatment in the setting of DA-resistant, aggressive, or metastatic prolactinomas. The AME/ICCE position statement suggests surgery should be undertaken if visual impairment is not reversed by DA therapy within 2 weeks [103]. Other treatments used in the multimodality management of these tumors are further discussed in the next section.

Skull base repair and tumor debulking in setting of CSF rhinorrhea (spontaneous or DA-induced): This has been considered mandatory by some experts [76], although published series have indicated that spontaneous resolution may occur even with ongoing DA use [77].

#### 3.5.2. Postoperative Evaluation

There is no consensus regarding optimal postoperative follow-up of prolactinoma patients. Some surgical series provides no data regarding the timing of the postoperative prolactin measurement. The Pituitary Society was unable to reach a consensus on a statement that: “Serum prolactin should be measured post-operatively (on postoperative day 1 or 2) to evaluate for remission” [113]. Ikeda et al. measured prolactin at days 7–10 and 6 months postoperatively [73]. Given the lack of data regarding optimal timing, it would appear reasonable to measure prolactin along with cortisol on postoperative day 1 and prolactin combined with serum Na and full pituitary function on day 7. Where visual fields have been abnormal, these should be repeated at 1–2 weeks post-operatively. The standard timing of the post-operative MRI scan is at 3 months [103], though notably, some series did not routinely undertake post-operative imaging [114]. When to resume DA therapy in patients not achieving a surgical remission is a clinical decision, based not only on serum prolactin concentrations but also on the functional outcome. In situations where a markedly elevated prolactin is reduced to less than twofold elevated in a pre-menopausal aged woman, for example, the fall may be sufficient for resumption of menses. Ongoing clinical and biochemical follow-up is required, complemented by repeat neuroimaging if the prolactin increases again after an initial decline.

### 3.6. Dopamine Agonist Resistance, Aggressive Prolactinomas and Metastatic Prolactin-Secreting PitNETs

Prolactinomas are defined as “aggressive” if they are radiologically invasive, cannot be cured by surgery, and have an unusually rapid growth rate despite optimal standard therapies (dopamine agonist therapy and surgery) [115,116]. A pituitary carcinoma, (now renamed “metastatic PitNET” in the 2022 WHO Classification of Pituitary Tumors [117]) is defined by the presence of metastases and is rare, representing 0.2% of PitNETs [116].

Dopamine agonist resistance does not necessarily imply that a prolactinoma is aggressive. The definition of DA resistance is variable throughout the literature [118]. The Endocrine Society guidelines define DA resistance as failure to achieve a normal prolactin on maximally tolerated doses of a DA and failure to achieve a 50% reduction in tumor size [96]. In an authoritative review by Maiter, DA resistance dose thresholds were defined as ≥15 mg daily for bromocriptine, ≥225 µg daily for quinagolide, and ≥2 mg weekly for cabergoline [46]. The estimated prevalence of DA resistance is 20–30% for bromocriptine and <10% (microadenomas) to 20% (macroadenomas) for cabergoline [46].

With cabergoline now being the DA of choice, recent series have focused on cabergoline resistant tumors. Eshkoli et al. have published their experience managing 26 cabergoline-resistant prolactinomas which they defined as a prolactin >three times the upper limit of normal on a cabergoline dose of >2 mg weekly [119]. Several of their cases would appear to fit the definition of an aggressive prolactinoma. Males were over-represented in the cohort, comprising 20/26 cases, and had larger tumors. The cabergoline was increased to a median dose of 3.5 mg weekly, 13 cases (50%) were managed with cabergoline alone, 9 received cabergoline plus surgery while 4 had cabergoline, surgery, and radiotherapy [119]. A normal prolactin was achieved in 7 (28%), while 14 (56%) fell to below 3x the upper limit of normal [119]. Resistance to cabergoline is also relative, and rarely 100% [46]. In a large series of 122 patients with macroadenomas, 83% achieved normal prolactin levels on ≤1.5 mg weekly. Further dose escalation up to 7 mg weekly resulted in normalization of prolactin in all but 6% of cases [28]. Therefore, the first step in the management of DA resistant prolactinoma is dose escalation. The study by Ono et al. used doses up to 12 mg weekly for resistant prolactinomas [47].

Recently there have been some studies regarding the use of DA sensitizing agents. Anastrozole, an aromatase inhibitor, was prescribed at a dose of 1 mg daily in 4 male patients with cabergoline-resistant macroprolactinomas whose prolactin failed to normalize on a cabergoline dose of >2 mg per week [120]. After a treatment duration of 15–56 months, serum prolactin fell by 44–97.4%, and there was a volumetric tumor reduction of between 24.5–68.7% compared to the best results achieved with cabergoline alone [120]. Metformin 1.0–2.5 g daily was studied as an add-on to unchanged doses of cabergoline in 10 cases of resistant prolactinoma for 120–180 days [121]. Mean prolactin did not change, but two participants had a ≥50% fall at a single time point [121].

The mainstay of adjunct medical therapy for the treatment of aggressive prolactinomas and metastatic PitNETs has been temozolomide [116]. Temozolomide is a lipophilic oral alkylating agent which can cross the blood brain barrier, commonly used in the treatment of glioblastoma [122]. It was first used in the treatment of PitNETs in 2006 [123], and since then aggressive prolactinomas have been the second most common PitNET treated [116]. It is administered at a dose of 150–200 mg/m^2^ for 5 consecutive days every 4 weeks. Most patients have been treated with a standard 6–12 month course, but longer courses of 2 years or more are now recommended [116]. Reduction in tumor size is observed in >50% of cases, and there is evidence of prolonged survival in responders [124]. The enzyme O6-Methylguanine-DNA Methyltransferase (MGMT) is a DNA repair enzyme whose expression on immunohistochemistry is negatively associated with temozolomide responsiveness [115]. Nevertheless, the ESE guidelines suggest trying temozolomide for 3–6 months regardless of the MGMT expression [115].

Other cytotoxic chemotherapeutic agents have had limited success in the management of aggressive prolactinomas [116]. Everolimus, an mTor inhibitor has been reported to induce a partial response in an aggressive prolactinoma [125].

Pasireotide is a somatostatin analogue that binds to the type 2 and type 5 somatostatin receptors (SSTR2 and SSTR5), which may be expressed on prolactinomas with much lower mRNA levels than the dopamine 2 receptor [126]. While the initial in vitro studies showed minimal prolactin inhibition using pasireotide [126], there have been case reports of impressive responses both in terms of prolactin level and tumor shrinkage [127,128]. The responsivity is related to high SSTR5 expression which occurs in only a small percentage (3/21) of prolactinomas [129].

Peptide receptor radionuclide therapy (PRRT) has been used in a small number of patients with aggressive prolactin secreting tumors [116]. Giuffrida et al. described 2 aggressive pituitary tumors (2 prolactinomas) treated with ^111^Ind-DTPA-octreotide (n = 1) an ^177^Lu-DOTATOC (n = 2). One of the aggressive prolactinomas responded with tumor shrinkage [130]. They also reviewed the response of 10 other pituitary tumors in the broader literature (only one prolactinoma and included one metastatic neuroendocrine tumor of ileal origin) to PRRT. Clinical response was observed in 4/13 patients and 1/3 prolactinomas [130]. Previous treatment with temozolomide was associated with absence of response to PRRT [130].

Lapatinib is a tyrosine kinase inhibitor (TKI) targeting the epidermal growth factor receptors ErbB1 and ErbB2, also known as HER2, which is approved for use in HER2 positive breast cancer. This was trialed in 4 patients at a dose of 1250 mg daily for 6 months [131]. No patient reached the primary end point which was a 40% reduction in any tumor diameter, three patients showed stable disease, while the 4th patient who has metastatic prolactin-secreting carcinoma exhibited progressive disease. Only one of the four patients had a fall in serum prolactin. The authors concluded that lapatinib may control tumor growth and reduce prolactin in selected patients [131].

Immunotherapy for aggressive prolactinomas has been tried on limited occasions. One case treated with nivolumab and ipilimumab had no response with evidence of progressive disease characterized by increasing hyperprolactinemia and tumor growth [132]. More recently Goichot and colleagues reported a prolactin secreting metastatic PitNET previously treated with repeated surgery and radiotherapy, followed by temozolomide which showed complete remission following treatment with nivolumab and ipilimumab after 2 years of follow up [133]. A phase 2 trial of nivolumab and ipilimumab is now underway in the USA.

Radiotherapy and stereotactic radiosurgery are established components of the multimodality treatment program for aggressive prolactinomas and metastatic PitNETs [116,134]. A recent systematic review evaluating the safety and efficacy of stereotactic radiosurgery for secretory PitNETs found a 93% rate of tumor control, 28% biochemical remission rate, and 12% rate of new onset hypopituitarism [135]. Withholding of anti-secretory medication is recommended for 4–12 weeks prior to the radiosurgery, although the evidence that dopamine agonists reduce the effectiveness of the treatment is equivocal [134]. A stereotactic radiosurgery approach is preferred to stereotactic fractionated radiotherapy unless the tumor is within 3 mm of the optic nerves of larger than 3–4 cm in diameter [134].

## 4. Conclusions

Prolactinomas are biologically diverse tumors, ranging from small indolent microadenomas to large, invasive macroadenomas and on rare occasions aggressive metastatic cancers. They are the most sensitive PitNET to medical therapy, and treatment with the dopamine agonist cabergoline remains a very appropriate and reliable modality for the majority of patients. Advances in pituitary surgery have now reached the stage where this is a valid first line treatment option for patients with microadenomas, non-invasive macroadenomas, and women seeking pregnancy, with the caveat that the surgery is performed by a specialist high volume pituitary neurosurgeon. Surgery remains appropriate for people with DA resistance or DA intolerance. Multimodality treatment is required for patients with aggressive prolactinomas and metastatic PitNETs, emphasizing the importance of pituitary tumors being managed via a multidisciplinary team.

## Data Availability

Data sharing is not applicable to this article as no data sets were generated or analyzed during the current study.

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
