# Peer review of "Treatment of Prolactinoma"

_medicina, 2022, doi:10.3390/medicina58081095_

Round 1
Reviewer 1 Report
approved
Reviewer 2 Report
Editing has been completed according to the requests
Reviewer 3 Report
The authors had appropriately addressed the reviewers' comments
This manuscript is a resubmission of an earlier submission. The following is a list of the peer review reports and author responses from that submission.
Round 1
Reviewer 1 Report
The article provides an interesting overview of the clinical management of prolactinoma, highlighting the emerging role of surgery and also providing an interesting insight on the treatment of aggressive forms. Just few minor comments:
- Paragraph 2:
- please consider to add few sentences about conditions in which PRL levels may be misleading. For example, in case of a giant macroadenoma with mildly elevated PRL ( es <100 ng/mL) , the elevation might be due to stalk compression of a non-functioning adenoma or to the so called “hook effect” in PRL measurement; if in doubt, the PRL must be measured with serial dilution of the sample if not routinely performed by the laboratory.
- please consider to add that measurement of prolactin, unless clearly elevated, should be preferably measured at rest, 15-20 min after the insertion of the i.v catheter, to avoid stress-related rise of PRL.
-line 56: galactorrhoea may also occur in males
-lines 63 and 69: the definition of “minor increase” and “moderate” hyperprolactinemia is vague. Please possibly define PRL cut-off for these definitions and/or provide references about this classification.
- line 72: please provide references about the definition of “poorly” functioning adenomas
- line86: the reason for a difference in size (micro in women and macro in men) is not fully explained by earlier diagnosis in women since, as also stated by the authors, microadenomas rarely grow to macro. Please consider to remove this sentence as may be confusing.
- Paragraph 3.1.1. The paragraph does not state the maximum dose of cabergoline unlike the other drugs, even if the topic is discussed later in the text. Please consider to add the maximum dose even in this paragraph
- Paragraph 3.3.1.
-line 289 We suggest against the measurement of PRL during pregnancy, as also stated by the latest ESE guidelines. Moreover, please specify that surgery should be considered in case of apoplexy when associated to visual deterioration ( Luger et al. ESE Clinical Practice Guideline on functioning and nonfunctioning pituitary adenomas in pregnancy. Eur J Endocrinol. 2021) .
- Paragraph 3.3.2. Line 303. Please consider to add “rarely” in the sentence “DA may cause psychosis…”
- Paragraph 3.4: A crucial element in considering withdrawal of DA is the menopausal state of the patient. Several evidence suggest that DA may be discontinued in post-menopausal women with microPRL while caution must be used in pre-menopausal women or patients with macroPRL. Please add considerations about the menopausal state in the paragraph.
- Line 381: Change the number of paragraph of suggested indication for surgery in 3.5.1 instead of 3.6.1
- For further indications on DA withdrawal and surgery indications Italian Association of Clinical Endocrinologists (AME) and International Chapter of Clinical Endocrinology (ICCE) have recently published a joined status on prolactinoma management ( Position statement for clinical practice: prolactin-secreting tumors. Eur J Endocrinol. 2022)
- Paragraph 3.6. Please consider to add PRRT among experimental treatments for aggressive PitNETs, although experience in this setting is currently limited ( Giuffrida G, et al . Peptide receptor radionuclide therapy for aggressive pituitary tumors: a monocentric experience. Endocr Connect. 2019)
- There are a few typos ( line 67 maybe is “which interrupts” instead of “with”; line 479 maybe is “selected patients” instead of “select”)
Reviewer 2 Report
In this wide review the Authors describe current multimodal management of prolactinomas. All the treatment modalities have been reviewed, from medical therapy to surgery.
The Authors went through diagnosis of pituitary prolactinomas, management by means of Dopamine Agonists (including a broad scope analysis of drug choice, doses, and adverse effects), surgery, and small insight on adjuvant therapy such as chemotherapy, immunotherapy and radiotherapy, including stereotactic radiosurgery.
The literature review and the clinical informationare up to date throughout the manuscript.
In the surgical section, the Authors clearly describe the historical approach to these pituitary tumors. Latest surgical indications for microprolactinomas, macroprolactinomas and DA-resistant, aggressive and metastatic prolactinomas are provided. Indication for surgical repair of CSF leak following tumor shrinkage after DA therapy are also discussed. Surgical complications have been analyzed as well. Accordingly, the Authors report a dramatic complication rate drop when surgery is performed in third referral centers with a dedicated pituitary unit. Similarly, remission rate is higher in those centers. In my opinion, these results should be highlighted even more.
DA withdrawal in case of biochemical and neuroradiological remission is discussed in a specific paragraph (i.e. paragraph 3.4), but there are no indications about DA withdrawal after surgery. I would suggest to add some data on this topic.
Furthermore, there are no indications regarding the timing of imaging or visual assessment after surgery. This is still debated and it could be an interesting addition to the paper.
Reviewer 3 Report
The authors should really shorten the manuscript and reduce the significant description of medical treatment side effects.
There is no novel consideration.
Pros and cons of the treatment needs to be addressed.
Literature and change trough time can help.